# New Insights on the Integrated Management of Plant Diseases by RNA Strategies: Mycoviruses and RNA Interference

**DOI:** 10.3390/ijms23169236

**Published:** 2022-08-17

**Authors:** Irene Teresa Bocos-Asenjo, Jonatan Niño-Sánchez, Mireille Ginésy, Julio Javier Diez

**Affiliations:** 1Department of Plant Production and Forest Resources, University of Valladolid, 34004 Palencia, Spain; 2iuFOR-Sustainable Forest Management Research Institute, University of Valladolid-INIA, 34004 Palencia, Spain

**Keywords:** plant protection, mycovirus, hypovirulence, RNAi, host-induced gene silencing (HIGS), spray-induced gene silencing (SIGS)

## Abstract

RNA-based strategies for plant disease management offer an attractive alternative to agrochemicals that negatively impact human and ecosystem health and lead to pathogen resistance. There has been recent interest in using mycoviruses for fungal disease control after it was discovered that some cause hypovirulence in fungal pathogens, which refers to a decline in the ability of a pathogen to cause disease. *Cryphonectria parasitica*, the causal agent of chestnut blight, has set an ideal model of management through the release of hypovirulent strains. However, mycovirus-based management of plant diseases is still restricted by limited approaches to search for viruses causing hypovirulence and the lack of protocols allowing effective and systemic virus infection in pathogens. RNA interference (RNAi), the eukaryotic cell system that recognizes RNA sequences and specifically degrades them, represents a promising. RNA-based disease management method. The natural occurrence of cross-kingdom RNAi provides a basis for host-induced gene silencing, while the ability of most pathogens to uptake exogenous small RNAs enables the use of spray-induced gene silencing techniques. This review describes the mechanisms behind and the potential of two RNA-based strategies, mycoviruses and RNAi, for plant disease management. Successful applications are discussed, as well as the research gaps and limitations that remain to be addressed.

## 1. Overview

Plant species, including agricultural, horticultural, and forestry crops, are constantly threatened by pests and pathogens. It is estimated that around 30% of crops worldwide are lost due to these harmful organisms [1]. Chemical fungicides and pesticides can release environmentally hazardous residues into the soil and groundwater and may cause unintended effects on non-target organisms. Large quantities of these products in the soil lead to the degradation of the microbial biodiversity of the soil, which causes a reduction in the enzymatic activity, thereby affecting nutrient availability, and a decrease in the biological nitrogen fixation, thus reducing soil fertility and impairing plant growth [2]. Furthermore, the overuse of chemical fungicides and pesticides has led to the development of resistance in some pathogens, making them even more difficult to control [3]. In addition, agrochemicals are sometimes ineffective in controlling diseases, which is often the case with forest pathogens. With climate change, plants are subjected to stressful conditions, making them more susceptible to infection, a problem exacerbated by the introduction of pathogenic organisms where they previously were not a threat as a result of rapid globalization [4]. Therefore, the need to develop new, environmentally friendly, and efficient alternative methods for plant disease control is pressing. In this context, taking advantage of what naturally occurs in the environment can offer solutions for the management of plant pathogens.

Mycoviruses are viruses, mostly harboring RNA genomes, that infect fungi. Some of them have been reported to decrease the virulence of their fungal host, a phenomenon called hypovirulence. This has opened up the possibility of using hypovirulence-inducing viruses to control plant pathogenic fungi. Mycoviruses commonly occur in fungi; however, only a few of them are deleterious to their hosts. This, coupled with the fact that mycoviruses are very difficult to transmit between fungi (as they are only transmitted intracellularly), limits their use as control agents. Furthermore, this system is only useful for the control of plant diseases caused by fungi. Nevertheless, with the advent of new sequencing techniques, numerous novel mycoviruses are being identified in a multitude of fungi, offering new possibilities in this field [5].

RNA interference (RNAi) is a gene silencing mechanism conserved among eukaryotes that regulates gene expression through the degradation of messenger RNA (mRNA) by the interaction of double-stranded RNA (dsRNA) molecules. Two strategies have been employed to exploit this mechanism for the control of plant pathogens: host-induced gene silencing (HIGS), which is based on the transgenic expression of specific interfering RNAs in the plant to confer resistance to a target pathogen, and spray-induced gene silencing (SIGS), where the topical application of specific dsRNA molecules degrades target genes in plant pathogens. The specificity of RNAi-based methods makes them a promising alternative to the use of conventional fungicides and pesticides. There is, however, greater interest in non-genetically modified (GM) exogenous approaches, such as SIGS, in light of some aspects of HIGS, especially the issues associated with public acceptance of genetically modified organisms (GMOs). In this work, we aim to review and discuss the use of RNA-based methods for plant disease management, focusing on mycoviruses and new RNAi-based approaches, especially SIGS, and the aspects to be considered to make them a viable option in the control and management of plant diseases. 

## 2. Mycoviruses: A Natural Source of Fungal Hypovirulence

### 2.1. Mycoviruses at a Glance

Mycoviruses are viruses that infect and replicate in fungi [6]. First described in 1962 [7], they are now known to be ubiquitous in numerous and diverse fungal taxa [8]. Like other viruses, mycoviruses require living cells of other organisms to replicate. RNA mycoviruses lack an extracellular route of infection in their life cycle, meaning that they have no natural vectors and rely solely on intracellular pathways via cell division, sporulation, and cell fusion (anastomosis) for transmission [9], a distinct characteristic that makes them unique. However, a DNA virus from the fungus *Sclerotinia sclerotiorum* named *Sclerotinia sclerotiorum* hypovirulence-associated virus 1 (SsHADV-1) is capable of extracellular transmission [10,11], notably via an insect vector [12]. Recently, further mycoviruses with this type of genome have been discovered [13,14], although their extracellular transmission has not yet been proven.

Viruses mostly store and replicate their genomes as RNA by encoding RNA-dependent RNA polymerases (RdRps). During viral infections, RdRps are involved in RNA template selection, RNA synthesis, and viral RNA preservation; hence, they are essential for virus survival [15]. According to the International Committee on Taxonomy of Viruses 2020 report, mycoviruses are classified into 23 different families, and more than 94% of mycoviruses have RNA genomes [16]. Most fungal viruses contain linear double-stranded RNA (dsRNA) genomes packaged in spherical particles. About 30% of mycoviruses have positive-sense single-stranded RNA (+ssRNA) genomes, while very few have negative-sense single-stranded RNA (−ssRNA) genomes. *Sclerotinia sclerotiorum* negative-strand RNA virus 1 (SsNSRV-1) was the first mycovirus characterized of this latter class [17], and subsequently, ssRNA mycoviruses have been identified in other fungi, such as *Botrytis cinerea* [18,19,20] and *Fusarium graminearum* [21]. Since most mycoviruses have RNA genomes and no extracellular phase, they simply are a fragment of RNA with self-replicating properties.

Most mycoviruses are considered cryptic: they remain latent and do not have any effect on their host [6,22]. Some have even established an endosymbiotic relationship with their hosts [8]. It is possible that mycoviruses infecting endophytic fungi (which are fungal endosymbionts of plants) help their host to adapt to rapidly changing or extreme environments through epigenetic effects [23]. Some might even be beneficial to the host plants, as observed with the *Curvularia* thermal tolerance virus that confers heat tolerance not only to the fungus it infects but also to the host plant of this latter [24]. However, there are also mycoviruses causing alterations and inducing abnormalities in their host, such as changes in fungal morphology, spore production, growth, pigmentation, virulence, and toxin production [25]. Such mycoviruses can stimulate fungal virulence, causing hypervirulence [26,27,28,29,30], or, on the contrary, reduce physical integrity or virulence in their fungal host, having a hypovirulent effect [31,32,33,34]. Mycoviruses causing hypovirulence are of particular interest as they could be used as virocontrol agents. It is unclear how viral genes interfere with fungal pathways, but some specific RNA-seq studies provide more insight into how mycoviruses regulate fungal genes at the transcriptome level [35]. 

In recent years, the emergence of high-throughput sequencing technologies has made possible the identification of an unexpected number of new mycoviruses among fungi (Table 1), a trend expected to continue in the future [5,36,37,38]. This stream of new findings is improving our understanding of the evolution and diversity of viruses. Fungal metatranscriptomics studies provide us a clue to the large number of mycoviruses that we have not identified yet, as demonstrated by Lee et al., who characterized the viromes of five plant pathogenic fungi and identified 66 previously unknown mycoviruses [39]. Myers et al. used double-stranded RNA-seq and total RNA-seq techniques to study 333 fungal specimens and found that 21.6% possessed one or more mycoviruses [40]. Similarly, 92 mycoviruses with different classes of genomes (62 novel) were identified in the mycovirome of 248 isolates of *B. cinerea* [20], 14 mycoviruses in four isolates of *Entoleuca* sp. [41], 10 mycoviruses in *Fusarium sacchari* and *Fusarium andiyazi* strains [42], and a large number of mycoviruses in *Rhizoctonia solani* isolates [43]. Deep sequencing of virus-derived small interfering RNA (resulting from antiviral RNA silencing) is also a useful method for detecting fungal RNA viruses, especially latent ones. For instance, Muñoz-Adalia et al. performed a first examination of the molecular antiviral response in *Fusarium circinatum*, characterizing some mitovirus [44]. Similarly, nine RNA viruses were identified in strains of four different species of the forest pathogen *Heretobasidion* [45], and two new RNA mycoviruses were discovered in a hypovirulent strain of *S. sclerotiorum* that already harbored the SsHADV-1 virus [46]. Therefore, high-throughput sequencing is a rapid and sensitive method to detect potential mycoviruses for biocontrol of fungal pathogens affecting horticultural, agricultural, agroforestry, or viticultural crops. 

Despite these ongoing major advances in mycovirus detection and identification, phenotypic evaluation remains a slow and laborious process. In addition, unknown information about host range, replication, infection, and transmission is indispensable for the use of mycoviruses as effective biological control agents.

### 2.2. How Mycoviruses Are Transmitted and the Obstacles They Face

Mycoviruses are transmitted vertically during sporogenesis and horizontally by cytoplasmic exchange during cell division, hyphal anastomosis, and mating [47]. The absence of extracellular transmission [9] has been attributed to the physical barrier formed by the fungal cell wall [48], and, indeed, cell wall removal has resulted in successful transmission of mycoviruses c [49,50,51].

Hyphal anastomosis between two fungal isolates is genetically controlled by a self/non-self recognition system called “vegetative incompatibility” (*vic*) [6,52]. It restricts the transmission between isolates belonging to different vegetative compatibility (*vc*) types: whereas vegetative compatible isolates can successfully establish hyphal anastomosis, the interaction of vegetative incompatible isolates triggers localized programmed cell death, which hampers sharing of genetic material [53]. As a result, mycoviruses have a narrow host range, generally limited to individuals of the same species and same *vc* type. However, phylogenetic evidence reveals occasional transmission across vegetatively incompatible strains [54,55]. The hypovirulence-inducing Cryphonectria hypovirus 1, until then found only in *Cryphonectria parasitica*, was identified in *Cryphonectria* sp. [56]. In strains of *Cryphonectria naterciae*, a new virus able to cross the species barrier to infect *Cryphonectria carpinicola* and *Cryphonectria radicalis* was recently identified [57]. The Botrytis porri RNA virus 1 (BpRV1), initially found in *Botrytis porri* strains, has also been found in *Botrytis squamosa* and *S. sclerotiorum*, suggesting that this mycovirus may have different hosts [58]. In addition, several studies have shown that vegetative compatibility is not an impossible obstacle to overcome. For example, Hamid et al. [59] found a novel mycovirus in *S. sclerotiorum* that was identified and described as the first +ssRNA mycovirus capable of spreading across the vegetative compatibility barrier [55,60,61]. Brusini and Robin found that although the transmission of CHV1 between *vic* strains was difficult in plates, it was much easier when the strains were inoculated in chestnuts [62]. *Vic* is apparently the result of biological and ecological factors, and even a model has been proposed in which mycoviruses can spread to vegetatively incompatible strains and to other fungal species through plant fungus-mediated pathways facilitated by plant viruses [63]. 

### 2.3. Hypovirulence as a Tool: Can We Copy Nature?

Mycoviruses that negatively affect their hosts are rare and interesting from the plant protection viewpoint. 

Mycoviruses inducing hypovirulence in their host are rare but interesting from a plant protection point of view since hypovirulent pathogens are unable to cause severe disease [25]. Since the discovery of hypovirulence, efforts have been made to exploit this phenomenon for the biocontrol of fungal diseases in agriculture and forestry [31,32,34,64,65]. 

The first hypovirulence-inducing mycovirus was found in *Cryphonectria parasitica*, a destructive fungal pathogen causing chestnut blight. Van Alfen et al. discovered a hypovirulent strain of *C. parasitica* [31], in which the presence of a virus-like genetic organization was later determined [66]. This mycovirus, called Cryphonectria hypovirus 1 (CHV-1), reduced the virulence of the pathogenic fungus in chestnut trees. Non-infected (virulent) strains penetrate and destroy the bark and cambium, causing wilting and death of the host, whereas CHV-1-infected (hypovirulent) strains usually produce superficial cankers that eventually heal without compromising the host’s survival [67]. The possibility of hypovirulence transmission to other strains of *C. parasitica* was exploited to use CHV-1 as a biological control agent [64,68]. Later, various mycoviruses belonging to different families have been described in *C. parasitica*, the most important for the biological control of chestnut blight being four species belonging to the family hypoviridae (+ssRNA): CHV-1 and CHV-2 in Europe; CHV-3 and CHV-4 in North America [69]. CHV-1 has received the most attention due to its success in controlling chestnut blight in Europe [70]. The CHV-1 genomic RNA is 12.7 kb in size, excluding a poly(A) tract with two contiguous open reading frames (ORFs) A and B [6]. The molecular basis of this fungus-mycovirus interaction is still up to debate. Segers et al. proposed that CHV-1 suppresses fungal RNA silencing (a host-derived antiviral mechanism), preventing the fungus from defending itself against the virus and causing hypovirulence [71]. It has also been proposed that the virus is able to inhibit the expression of a fungal gene encoding a hydrophobin, a protein required for the eruption of fungal fruiting bodies through the host cortex, thereby decreasing *C. parasitica* virulence [72,73]. RNA-seq studies aiming at identifying differences between virulent and hypovirulent strains of *C. parasitica* have shown that the presence of CHV-1 affected a multitude of metabolic pathways: genes related to fungal metabolites, signaling pathways, virulence, and antiviral RNA silencing [35]. However, RNA-seq does not allow to determine whether these alterations are due to the particular effect of CHV-1 or simply to a defense response of the host organism. Therefore, the effects of hypovirulence caused by CHV-1 infection in *C. parasitica* are well-known [70], but the underlying molecular mechanisms remain uncertain and may be related to post-transcriptional interactions such as gene silencing or changes in the metabolism [35]. 

Following the discovery of CHV-1, mycovirus-mediated hypovirulence in fungal pathogens of plants ranging from grass to trees has been reported frequently (Table 2). Recently, Pestalotiopsis theae chrysovirus-1 (PtCV-1), a mycovirus found in a tea pathogen that converts its host into a non-pathogenic endophyte of tea leaves, has been identified [74]. In *Botrytis cinerea*, the causal agent of gray mold disease on more than 1400 plant species, including many economically important crops, the Botrytis cinerea partitivirus 2 (BcPV-2) was found in hypovirulent isolates in apple, cucumber, oilseed rape, strawberry, table grapes, tobacco, and tomato, suggesting that this mycovirus might be responsible of the hypovirulence [75]. The Colletotrichum liriopes partitivirus 1 (ClPV-1) could reduce the virulence and conidia production of *Colletotrichum liriopes*, a fungus causing leaf anthracnose in many plants [76]. In forestry, in addition to the mycovirus model CHV-1 affecting *C. parasitica*, several mycoviruses triggering hypovirulence have been described in fungal pathogens. Hypovirulent strains of *Ophiostoma novo-ulmi*, the causing agent of the devastating Dutch Elm Disease in *Ulmus* spp., harbor a dsRNA virus called Ophiostoma novo-ulmi mitovirus (OnuMV) that reduces fungal growth in wounds caused by some beetles (vector) when feeding [77]. Another example of hypovirulence was found in *Botryosphaeria dothidea*, a pathogen of global importance for woody plant health, which infects a broad range of hosts. Three dsRNA mycoviruses producing an attenuated fungal growth and virulence have been discovered in hypovirulent isolates of *B. dothidea*: Botryosphaeria dothidea chrysovirus 1 (BdCV1) and Botryosphaeria dothidea partitivirus 1 (BdPV-1) [78] and more recently Bipolaris maydis botybirnavirus 1 strain BdEW220 (BmBRV-1-BdEW220) [79]. Recently, the hypovirulence mechanism in this pathogen was investigated by analyzing full genome sequences of one virulent strain infected by BdPV-1, one attenuated strain infected by BdCV-1, and one virus-free control strain. The study revealed that the interaction of *B. dothidea* and mycoviruses involves the coupled action of the antiviral gene silencing pathway and micro-like RNAs-mediated regulation of target gene mRNA expression in *B. dothidea* [80]. However, the mechanistic details remain to be elucidated. Some authors have studied the fungal antiviral mechanism in the presence of mycoviruses and surprisingly have shown that virus-derived small interfering RNAs (antiviral function) could target specific fungal host genes, thus inducing their silencing [81,82]. Mycovirus-associated hypovirulence could therefore be modulated by RNAi antiviral responses. Despite efforts to clarify the mechanisms of hypovirulence, these are not fully known yet and may actually be specific to the lifestyle of each virus.

In a nutshell, all known mycoviruses causing hypovirulence in fungal hosts harbor dsRNA or ssRNA genomes, except for one that contains DNA [32,33] and include representatives of the *Totiviridae*, *Chrysoviridae*, *Hypoviridae*, *Narnaviridae*, and *Reoviridae* families [8,83]. A large number of mycoviruses have been described; however, the ones causing hypovirulence are very few and are not found in all phytopathogenic fungi.
ijms-23-09236-t002_Table 2Table 2Mycoviruses described in the literature that trigger hypovirulence in the fungal host.Mycovirus *GenomeFungal HostMycovirus FamilyHost PlantFungal DiseaseReferenceCHV-1+ssRNA*Cryphonectria parasitica**Hypoviridae**Castanea sativa*Chestnut blight[84]CHV-2+ssRNA[85]CHV-3+ssRNA[86]OnuMV+ssRNA*Ophiostoma novo-ulmi**Narnaviridae**Ulmus* spp.Dutch elm disease[77]SsMV-1/HC025+ssRNA*Slerotinia sclerotiorum**Narnaviridae**Glycine max*, *Brassica napus*, *Lupinus angustifolius*, *Pisum sativum*White mold[87]SsHADV-1ssDNA*Genomoviridae*[32]SsHV-1+ssRNA*Hypoviridae*[88]SsHV-2+ssRNA*Hypoviridae*[89]SmEV-1+ssRNA*Sclerotinia minor**Endornaviridae**Lactuca sativa*, *Arachis hypogaea*, *Brassica rapa*, *Brassica napus*, sunflowerSclerotinia blight[90]AaCV-1dsRNA*Alternaria alternata**Chrysoviridae*Herbaceous annual plants, ornamental plants, and trees (citrus, apple, etc.)Leaf spots, rots, and blights[91]AaHV-1+ssRNA*Hypoviridae*[92]FgV-ch9dsRNA*Fusarium graminearum**Chrysoviridae*Small-grain cereals (wheat and barley)Fusarium head blight (FHB)[93]FgHV-2+ssRNA*Hypoviridae*[94]FodV-1dsRNA*Fusarium oxysporum f.* sp *.dianthi**Chrysoviridae**Dianthus caryophyllus*Carnation disease[95]BcMV-1+ssRNA*Botrytis cinerea**Narnaviridae*Vegetables and small fruit crops (tomato, raspberry, grape, strawberry, blueberry, apple, and pear)Gray mold disease[96]RnMBV-1dsRNA*Rosellinia necatrix**Megabirnaviridae*Fruit trees (apples, apricots, avocados, cassava, strawberries, pears, citruses, and Narcissus)Rosellinia root rot[97]PtCV-1dsRNA*Pestalotiopsis theae**Chrysoviridae**Camelia sinensis*Thea blight[74]BdCV-1dsRNA*Botryosphaeria dothidea**Chrysoviridae**Pyrus pyrifolia*Pear ring spot[78]BdPV-1dsRNA*Partitiviridae*BmBRV-1-BdEW220dsRNA*Botybirnaviridae*[79]* CHV-1: *Cryphonectria hypovirus* 1; CHV-2: *Cryphonectria hypovirus* 2; CHV-3: *Cryphonectria hypovirus* 3; OnuMV: *Ophiostoma novo-ulmi mitovirus*; SsMV-1: *Sclerotinia sclerotiorum mitovirus* 1 strain HC025; SsHADV-1: *Sclerotinia sclerotiorum* hypovirulence-associated DNA virus 1; SsHV-1: *Sclerotinia sclerotiorum hypovirus* 1; SsHV-2: *Sclerotinia sclerotiorum* hypovirus 2; SmEV-1: *Sclerotinia minor endornavirus* 1; AaCV-1: *Alternaria alternata chrysovirus* 1; AaHV-1: *Alternaria alternata hypovirus* 1; FgV-ch9: *Fusarium graminearum* virus China 9; FgHV-2: *Fusarium graminearum hypovirus* 2; FodV-1: *Fusarium oxysporum* f. sp. dianthi virus 1; BcMV-1: *Botrytis cinerea mitovirus* 1; RnMBV-1: *Rosellinia necatrix megabirnavirus* 1; PtCV-1: *Pestalotiopsis theae chrysovirus*-1; BdCV-1: *Botryosphaeria dothidea chrysovirus* 1; BdPV–1: *Botryosphaeria dothidea partitivirus* 1; BmBRV-1-BdEW220: *Bipolaris maydis botybirnavirus* 1 strain *Botryosphaeria dothidea* EW220. 


### 2.4. Virocontrol: Lessons Learned and Challenges Ahead

Hypovirulence-associated mycoviruses are biologically interesting for the control of diseases caused by their fungal hosts. Nevertheless, there are still many aspects that need attention and question their potential as biological control agents. Indeed, for efficient disease control, a mycovirus needs to: (a) cause hypovirulence in the host; (b) have a high vertical transmission rate through spores; (c) skip host mechanisms to control the spread and infection of viruses. In fungi, RNAi is known to play a critical role in antiviral defense. Indeed, studies with disruptive mutants of essential genes of the RNAi machinery show a severely weakened fungal phenotype [98]. However, some viruses can escape these defenses, such as mitoviruses that, by accumulating in mitochondria, evade RNAi defense [99]. Other mycoviruses counteract this host defense response by encoding RNA silencing suppressors [71]. In addition, fungi may have other mechanisms to control the propagation of mycoviruses, such as the production of secondary metabolites [100], although little information is yet available on mycovirus-fungal interactions. 

In addition, to enable effective horizontal transmission, the diversity of host vegetative compatibility types in the area of treatment must be low, and the hypovirulent strains must be compatible with the dominant *vc* types in the area of treatment. In the case of CHV-1 and *C. parasitica*, these requirements are fulfilled in Europe (hypovirulent effect, low diversity of *vc* types, and hypovirulent strains compatible with the dominant *vc* types); however, chestnut blight is so far the only disease that has been effectively controlled with an RNA mycovirus in the field [64,101,102,103]. 

One of the main challenges posed by virocontrol nowadays is the search for mycoviruses with a hypovirulent effect. Despite high-throughput sequencing technologies allowing rapid and massive identification of viruses and the recent and updated sequence banks, no known bioinformatics model predicts hypovirulence. For example, the Serratus platform (https://serratus.io/ accessed on 16 August 2022) is a newly developed cloud computing infrastructure for ultra-high-throughput sequence alignment that facilitates the discovery of new viruses [104]. Serratus enables inexpensive processing of massive data sets that are freely available but usually under-exploited. However, mycoviruses have to be finally tested by inoculation assays in the laboratory in order to ascertain their hypovirulence. 

The constraints of their interspecies and even intraspecies transmission also pose a challenge to their applicability, even once hypovirulence has been discovered, because both high horizontal and vertical transmission rates are necessary in order to ensure the durability of disease control and transmission of viruses to field strains.

Thus, future efforts should focus on: finding and readily identifying new mycoviruses with hypovirulent effects in more plant pathogenic species, developing techniques to overcome their transmission limitations, and enabling easy application in the field. 

## 3. RNA Interference (RNAi): The Targeted Management for Plant Disease Control

### 3.1. An Overview of RNAi

RNA interference (RNAi) was first described by Fire et al. in the nematode *Caenorhabditis elegans*, which earned the authors Andrew Fire and Craig Mello the Nobel Prize for Medicine in 2006 [105]. However, this phenomenon had actually already been observed in *C. elegans* [106]. Moreover, evidence of RNAi, then referred to as post-transcriptional gene silencing or quelling, had also been previously found in tobacco and petunia plants [107,108] and in the fungus *Neurospora crassa* [109], respectively. Since then, RNAi has been discovered in insects [110] and mammals [111] as well. RNAi is present in almost all eukaryotes examined, and so far, only a few eukaryotes, such as *Saccharomyces cerevisiae* and *Ustilago maydis*, have been found to be insensitive to RNAi silencing, as they have lost some key RNAi enzymes [112]. Key proteins involved in RNAi are highly conserved in different organisms, suggesting that the last common ancestors of modern eukaryotes possessed an RNAi mechanism. This conserved protein machinery in eukaryotes appears to be constructed from proteins involved in DNA repair and RNA processing in ancestral archaea, bacteria, and phages [113]. 

RNAi is initiated by small interfering RNA (siRNA) duplexes, which are 19–25 nucleotide-long double-stranded RNA molecules produced by DICER-mediated cleavage of longer double-stranded RNAs (dsRNA). Then, a single-stranded guide RNA is incorporated into the RNA-induced silencing complex (RISC), resulting in the endonucleolytic cleavage of the complementary mRNA, thereby regulating gene target expression [4,114,115]. Since RNAi-mediated silencing depends on the recognition of complementary sequences, it is a specific process, although strict homology between the mRNA and the complete siRNA sequence is not required, only a stretch of about 8 bp, known as seed sequence [116].

RISC can potentially down-regulate any mRNA with perfect base complementarity to the seed region of the guide strand [117]. The RNAi effect can be amplified due to cellular RNA-dependent RNA polymerases (RdRPs), which trigger a transitive generation of secondary siRNAs (by amplification of the antisense strand of the mRNA target). RdRPs are most likely responsible for the strong RNAi effect in most eukaryotic organisms [4,118]. 

RNAi is believed to have evolved as a defense mechanism against viruses and transposable elements, which can affect the integrity of genomic DNA [119]. Exposure to foreign genetic material (often dsRNA) triggers the gene silencing response to eliminate the invader [120]. In addition to protecting organisms against foreign nucleic acids, RNAi mediates host immune mechanisms [121,122], pathogen virulence [123], and host-pathogen communication [124,125] through microRNAs (miRNAs), which are small RNAs (sRNAs) naturally generated from specific genome-encoded precursors. 

RNAi machinery is not present in prokaryotes, but they have a functionally similar defense system to induce the inactivation of parasite genomes: they produce small non-coding RNAs that can up- or down-regulate mRNA stability and translation [126,127]. 

### 3.2. How Nature Works and How We Benefit from It: HIGS and SIGS

RNA molecules can move across the cellular boundaries between hosts and their pathogens, pests, and parasites and induce gene silencing through RNAi [128]. This communication between interacting organisms is called ‘cross-kingdom RNAi’ [129]. Several studies have also shown that some sRNAs are exchanged between different hosts, plants or animals, and their pathogens [124,130,131,132,133]. Cross-kingdom RNAi communication between plant hosts and pathogens is bidirectional [124]. Indeed, the pathogen *B. cinerea* transfers RNAi effectors into host plant cells to induce silencing of host immune response genes and achieve infection [130], while sRNAs from host plant cells are transferred to fungal cells, triggering RNAi [117], exemplifying cross-kingdom RNAi and sRNA trafficking between plants and fungi. Thus, naturally occurring trans-species sRNAs are used by pathogens to silence host mRNAs and by host plants to silence mRNAs from the pathogens [123,125,134]. The sRNAs delivery mechanisms from plants to pathogens are not fully understood. It has been shown that plants release extracellular vesicles containing the sRNAs for silencing fungal virulence genes [125,135]. Recently, it has also been suggested that sRNAs travel in association with protein complexes outside of the extracellular vesicles [136]. Trans-kingdom gene silencing between plants and pathogens also occurs with oomycetes. *Arabidopsis* plants were found to produce a pool of siRNAs contained in extracellular vesicles in response to *Phytophthora capsici* infection, leading to the impairment of pathogen development [134]. Reciprocally, *P. capsici* tried to suppress the RNAi response [134]. Interestingly, it has recently been shown that host mRNAs also travel inside extracellular vesicles and are biologically active when translated into the fungal pathogen cells [137].

#### 3.2.1. HIGS

Host-induced gene silencing (HIGS), a term coined by Nowara et al. [138], exploits cross-kingdom RNAi to control plant diseases by genetically modifying plants to express RNAs targeting pathogen genes, inducing gene silencing and conferring disease resistance to the plant [139]. Transgene expression of silencing RNAs offers long-lasting protection in plants and versatility to silence different genes and even different pathogens at the same time [140] and presents an alternative to the application of chemical insecticides and fungicides. 

The first report on HIGS for plant protection showed that transgenic *Arabidopsis* expressing dsRNAs targeting a nematode parasitism gene were resistant to root-knot nematode [128]. This was followed by work on insects: the growth of cotton bollworm larvae was successfully inhibited when fed leaves of transgenic plants expressing dsRNAs to silence their cytochrome P450 gene [141]; significant protection against western corn rootworm was achieved by *in planta* expression of gene-silencing dsRNAs [142]. In fungal pathogens, Nowara et al. used HIGS to successfully inhibit the development of the biotrophic powdery mildew fungus, *Blumeria graminis*, in genetically modified barley [138]. In oomycetes, transgenic potato plants expressing dsRNAs were resistant to *Phytophthora infestans* [143]. Since then, much research has focused on the use of HIGS for the control of important plant pathogenic fungi and oomycetes [124,144,145,146,147,148]. 

Over 170 studies on HIGS have been published so far [139]. However, there are only a limited number of HIGS-based products for crop protection on the market. The first product of this kind is SmartStax^®^ Pro (https://traits.bayer.com/corn/Pages/SmartStax-PRO.aspx accessed on 16 August 2022), which is a GM corn seed that deploys transgenic insecticidal proteins and RNAi to fight the corn rootworm [149]. This product is available to farmers in the United States (from 2022) and in Canada (from 2023); in Europe, it is authorized for all uses except cultivation due to the strong regulation of GMOs [150]. 

An increasing number of GM crops are being developed, risk assessed, and accepted by relevant international regulatory agencies [151]. The European Food Safety Authority concluded in a risk assessment that consuming RNAi products entails a low risk of interfering with gene expression in humans due to the many biological and physical barriers to overcome [152,153]. However, HIGS-based products are regulated as GMOs in the European Union and fall under the scope of Directive (EC) 2001/18 [154], which tightens requirements for the use of GMOs. If these products are intended for food or feed, then they fall under Regulation (EC) 1829/2003 [155], which prohibits their placing on the market and the authorization of GMOs for food use, thereby greatly limiting HIGS application. The lack of transformation protocols for some species, extensive capital requirements, and difficulty and time required to develop GM products, coupled with regulatory hurdles and political/public concerns, further restrict the use of HIGS as a plant disease management tool [156,157,158]. Recently, efforts have thus focused on the development of GM-free alternatives for the use of RNAi in crop protection, such as SIGS. 

#### 3.2.2. SIGS

RNAi response in pathogens induced by external RNAs uptaken from the environment, referred to as ‘environmental RNAi’, was first observed in *C. elegans* [159,160,161]. Environmental RNAi discovery has led to the development of spray-induced gene silencing (SIGS), a strategy for crop protection where artificially synthesized dsRNA molecules targeting susceptible pathogen genes are sprayed on plants’ surfaces and taken up by pathogens cells, resulting in disease reduction.

SIGS has also been shown to be effective against fungi. When Wang et al. externally applied dsRNAs synthesized in vitro targeting Dicer-like proteins in the pathogen *B. cinerea* to different post-harvest plant materials (including vegetables, fruits, and flowers), they found that all plants treated with dsRNAs developed much weaker disease symptoms [124]. Similarly, spraying detached barley leaves with a solution of dsRNAs targeting three genes of the ergosterol biosynthesis pathway in *F. graminearum* conferred strong resistance against this pathogen [162]. Koch et al. tested SIGS targeting the same genes in *Fusarium culmorum*, obtaining satisfactory results in reducing the growth of this fungus, which suggested that this technique had unprecedented potential for easy application against various fungal pathogens [163]. Koch et al. silenced ergosterol biosynthesis-related genes in *F. graminearum* testing HIGS and SIGS approaches [164]. They showed that SIGS was very effective against the pathogen and that dsRNAs designed to target several genes were more efficient than those targeting only one [164]. McLoughlin et al. developed a pipeline to identify pathogenicity-related genes from RNA-seq analysis of tolerant and susceptible *Brassica napus* leaves infected by *S. sclerotiorum*, which they used to identify and design dsRNA molecules targeting genes associated with reactive oxygen species [165]. The topical application of these dsRNAs on plants led to a significant reduction in lesion size following infection. Recently, many studies have sought to find targets for the development of RNAi-based fungicides for agronomically important pathogens [166,167].

SIGS technology can also be applied against oomycetes [168,169]. Interestingly, these studies showed that dsRNAs simultaneously targeting two genes reduced infection to a higher degree than dsRNAs targeting a single gene, as has been observed in fungi. This provides an important basis for the development of RNA-based anti-oomycete agents.

SIGS is likely to be technologically simple, safe for consumption, and socially accepted as it does not require the generation of GMOs. Moreover, it allows effective post-harvest control of fungi, which has not been well demonstrated with HIGS [124,170]. SIGS can simultaneously control various pathogens and is more practical for the control of pests and pathogens affecting multiple crops [171]. Furthermore, RNA degrades in a short time, and potential accumulation in the environment should not be dangerous since nucleic acids are already present in nature [172]. Therefore, RNAi-based pesticides represent an environmentally friendly alternative to agrochemical products and to solutions based on the genetic engineering of plants [173]. 

### 3.3. Implementation of SIGS in Plant Protection: Major Challenges

Some challenges remain to be addressed to enable the large-scale production and commercial use of RNAi-based biopesticides and biofungicides: (1) the uptake efficiency, (2) the target choice, (3) the protection duration and stability of dsRNA, as well as (4) the safety, (5) the environmental implications, and (6) the economic competitiveness of SIGS.

#### 3.3.1. Uptake Efficiency

The ability of the target pathogenic organisms to incorporate the dsRNA molecules The ability of the target pathogenic organisms to incorporate the dsRNA molecules is key to the success of this SIGS management strategy and determines its effectiveness. Recently, Qiao et al. assessed the uptake efficiency in different plant pathogenic fungi by applying fluorescein-labeled dsRNA and analyzing the fluorescent signal using a confocal microscope [174]. They established that not all pathogens are able to efficiently capture long and/or small dsRNAs and that this constitutes a determining factor in their control by RNAi. Indeed, a significant reduction in plant disease symptoms was observed with fungi that efficiently took up dsRNA targeting virulence-related genes (*Sclerotinia sclerotiorum*, *Rhizoctonia solani*, *Aspergillus niger*, and *Verticillium dahliae*). On the contrary, fungi that did not take up dsRNA, such as *Colletotrichum gloeosporioides*, did not experience a reduction in infection. The size of the dsRNA influences the uptake efficiency. Indeed, Höfle et al. found that molecules between 400 and 500 bp were most effective, while from 800 bp onwards, efficiency decreased, and molecules over 1500 bp had no silencing effect, suggesting they were not taken up by the fungus [175]. dsRNA molecules can either remain on the plant surface and be taken up by the fungus directly or be taken up by the plant, processed into siRNAs and then transferred to fungal cells [176]. Interestingly, the silencing effect was greater and longer lasting when the dsRNAs were first taken up by the plant rather than directly by the fungus. In addition, dsRNAs uptake was more effective when the plant surface was wounded beforehand [176]. Koch et al. found that dsRNAs were more readily taken up by hyphae in close contact with the plant and that hyphae specialized in colonizing plant tissue show a better dsRNAs uptake [162]. It is also known that sRNAs can spread within the plant via the phloem [177,178]. Betti et al. even showed that plant cells produce miRNAs that act as signals for plant-to-plant communication [179]. Consequently, each particular disease-causing pathogen must be studied individually to assess the possibility of SIGS as a management method. 

#### 3.3.2. Target Choice

The choice of effective target genes is essential for the success of the silencing and disease control strategy. First of all, appropriate candidate genes of the target pathogen that are vital for its growth and pathogenicity must be identified [180]. Secondly, recent studies have shown that not all genes are susceptible to silencing by RNAi; in fact, some genes are even refractory to RNAi [181]. Finally, the inability of RISC to unfold structured RNA implies that accessibility of the target site directly correlates with cleavage efficiency [182]. The efficiency of target gene silencing by RISC is fluctuating, and although this may be due to factors such as RISC assembly or activation, accessibility at the target site has been found to correlate directly with excision efficiency, as RISC is unable to unfold structured RNA [182]. 

As mentioned, several target pathogens, or several gene targets, can be silenced simultaneously. This is very useful as plant diseases are sometimes caused by a group of pathogens rather than by a single one. Silencing several genes or different regions of a gene at the same time can enhance the silencing effect and expand the target species [125].

dsRNA molecules must be designed to optimize silencing efficiency: taking into account the secondary structure of the selected target sequence since complex RNA structures can prevent base-pairing between sRNA and the target, which inhibits cleavage of mRNA by the RISC; avoid secondary structure formation in the guide RNA, which can significantly reduce silencing effect [183,184,185,186]. There are now available siRNA design tools that consider the accessibility of the target site, such as the RNAxs tool (http://rna.tbi.univie.ac.at/cgi-bin/RNAxs/RNAxs.cgi accessed on 16 August 2022) [187]. 

Since it relies on the recognition of specific sequences, SIGS can be very specific for the target pathogen, which is important for disease management and represents a clear advantage over non-selective chemical products [114]. However, the risk of off-target effects cannot be neglected, even with SIGS. The seed sequence or minimum homology needed to cause silencing is thought to be about 8 bp [1], which would then also be enough to cause off-target effects. However, other authors suggested that 11 contiguous nucleotides or 15 out of 19–25 complementarities can also cause off-target silencing [188]. Chen et al. tried to find out the minimum length of imperfectly matched dsRNAs to induce off-targets and observed differences between the more susceptible and less susceptible genes to silencing [189]. They found that dsRNAs with >80% sequence identity with target genes, with ≥16 bp segments of perfectly matched sequence, or with >26 bp segments of almost perfectly matched sequence with one or two mismatches scarcely distributed, triggered RNAi efficiently, which suggests that off-target effects correlate with mismatch rates between dsRNA and non-target mRNA. In the same study, high silencing rates in non-target insects were observed. Therefore, target genes with high homology in host plants or beneficial microbiota should be avoided, taking both sense and antisense strands (of the dsRNA designed) into consideration because either could serve as a guide RNA strand. Some website-based open-access programs provide useful sRNA blasting free tools for RNAi studies, notably for the design of RNAi constructs considering off-target effects and accessibility (Table 3).

#### 3.3.3. Protection Duration and dsRNA Stability

Several factors influence the duration of the silencing effect, limiting the use of SIGS as a control method. Several studies pointed out a short pathogen protection window post spray: 5–10 days, usually one week, with the efficacy of dsRNA decreasing over time [124,162,174,176,195]. The unstable nature of RNA in field conditions (ultraviolet light, oxygen, and temperature variations) is a major hurdle in adapting SIGS approaches for widespread applications [172]. Therefore, it is necessary to properly design dsRNA application strategies to: improve the uptake capacity of dsRNAs and increase their durability so that they are maintained long enough for uptake to occur [196]. Approaches developed for application in plants include inorganic nanoparticles as RNA carriers, such as: layered double hydroxides nanosheets (BioClay^TM^ when combined with dsRNA) [195,197,198], carbon nanotubes [199,200], carbon dots [201], a mix of chitosan, carbon quantum dots, and silica nanoparticles [202]; or gold nanoparticles [203,204], among others. Several nanoparticles have been studied to determine their optimal ζ potential for cell uptake [205] and some target peptide recognition motifs [206]. Another potential method for RNA delivery is the use of organic nanoparticles as carriers, e.g., packaging dsRNAs in liposomes- or extracellular vesicles-like structures using artificially synthesized phospholipid bilayers, in order to mimic the natural mechanism by which plants deliver their own sRNAs to fungal pathogens [125,207,208]. Naturally formed lipid nanoparticles isolated from plants also can deliver siRNAs as therapeutic agents [209,210]. The plant apoplast contains naturally occurring protein complexes associated with dsRNAs [136]; therefore, it has been reasoned that peptide-based carriers could be used for RNA delivery [211,212]. 

All these formulations can protect sRNA and also be taken up by pathogens. Carriers’ features can be optimized to ensure efficient delivery: particle size, carrier shape, dsRNA dose, carrier-dsRNA complexing ratio, and treatment time. It was found that sheet-like clay dsRNA nanoparticles up to 50 nm in diameter were easily internalized [213]. Further, Zhang et al. demonstrated that dsRNA can be delivered efficiently without cellular internalization, ensuring a slower cargo release [214]. 

In addition to inorganic and organic carriers, another delivery method being investigated is the use of vectors that have been genetically modified to produce dsRNAs. Niño-Sánchez et al. first demonstrated that silencing can be induced by the application of living or lysed bacteria engineered to produce biologically active dsRNAs targeting fungal genes [215]. Further development and improvement of methods to minimize the degradation of dsRNA molecules and to improve their stability and durability are needed to allow this SIGS to be used in the field.

#### 3.3.4. Technology Safety

sRNAs are normally present in all plant- and animal-derived food. In addition, sRNAs with sequences complementary to humans and animals have been found in crops widely consumed globally [216]. Some approved RNA-based crop protection traits have been used in the field for decades [217,218]. This long history of consumption of both endogenous and exogenous dsRNAs in food and feed seems to indicate that this technology has no negative health effects. Moreover, for these molecules to reach the target genes of their consumers, they would have to overcome numerous difficult biological barriers [219]. Thus, dsRNA crop treatments are a safer alternative for human and animal health than the use of agrochemicals and are less harmful to the environment. Although regulatory authorities have not yet established standard procedures for assessing dsRNA-based agricultural products, it is expected that the existing robust regulatory framework for small molecule agrochemicals could be applicable [219].

#### 3.3.5. Environmental Implications

Appropriate construction of silencing molecules should prevent unintended consequences on the environment, such as the silencing of non-target organisms. SIGS-based products should not leave residues in soil, water, or plants due to the non-permanent nature of dsRNA molecules, and even in case it would, the natural occurrence of RNA in the environment hints that it would not be a problem [172,220]. In contrast, traditional chemical pesticides and fungicides entail some concerns as they release residues that can remain in the environment for a long time and pose a serious threat to the environment and to living organisms, including humans. In addition, they generate resistance in the target pathogens, with the rate of emergence of pathogenic fungi resistant to the limited number of commonly used antifungal agents increasing at an alarming trend. This leads to an overuse of these harmful products, accumulation of residues, and ecological damage, and it also may have risks to human health [221]. Pathogens are less likely to develop resistance to RNAi-based fungicides because complete homology is unnecessary for effective silencing [222]. 

#### 3.3.6. Economic Competitiveness

RNAi gene silencing in vivo requires a large amount of dsRNA to be effective: 2–10 g of dsRNA per hectare, as predicted by Zotti et al. [223]. Fortunately, the costs of dsRNA production by NTP synthesis are becoming lower and lower (from $12,500 USD/g in 2008 to $60 USD/g in 2018 and less than $0.5 USD/g nowadays) [124,171,223]. Nonetheless, this method is not practical for large-scale production. Microbial-based dsRNA production methods, relying, for instance, on the *Escherichia coli* HT115(DE3) strain deficient in the enzyme that degrades dsRNAs in prokaryotes [160,224], costs approximately $4 USD/g [223]. Thus, large quantities of dsRNA can be produced with bacteria-based systems at a cost-effective price. Moreover, the application of living dsRNA-producing bacteria for gene silencing has been used in nematodes [160], insects [225], crustaceans [226], mammalian cells [227], and filamentous fungi [215]. However, in order to avoid bacterial release into the environment, it is necessary to purify the RNA, which can make the process more expensive and limit the amount of dsRNAs produced [26,228,229,230,231,232]. 

The multinational companies Monsanto, which has developed a line of RNA-based compounds known as “BioDirect”; Syngenta (Chem-China) and Bayer Crop-Science, which has now acquired Monsanto; are working on the development of dsRNA compounds. BioDirect (dsRNA-based) has been developed to control *Varroa destructor* mites in honeybees. It has been submitted by Bayer for registration in 2019, being the first submission to the U.S. Environmental Protection Agency for an exogenously applied dsRNA biopesticidal active ingredient in the industry [233].

Start-up companies are also involved in the task of exploiting biotechnology to offer cost-effective solutions to the challenges posed by SIGS. For example, GreenLight Biosciences (https://www.greenlightbiosciences.com/ accessed on 16 August 2022), RNAissance (https://www.rnaissanceag.net/ accessed on 16 August 2022), and Genolution (http://genolution.co.kr/ accessed on 16 August 2022) are working on low-cost dsRNA synthesis. RNAissance, AgroSpheres (https://www.agrospheres.com/ accessed on 16 August 2022), Nanosur (http://www.nanosur.com/ accessed on 16 August 2022), and Agrisome (TrilliumAg) (http://www.trilliumag.com/ accessed on 16 August 2022) work on the protection of RNA molecules from degradation [171]. Greenlight Biosciences offers a very efficient and economical alternative for dsRNA production, as it achieves the same bioactivity and specificity as in vitro transcription for a very low price (<$0.50 USD/g) [233]. This technology is based on a cell-free production method, in which, by decomposing a low-cost RNA substrate, it constructs the desired dsRNA molecules with a DNA template, enzymes, and an affordable energy source [234]. Greenlight biosciences have tested its technology in field trials against the Colorado Potato Beetle, providing protection against this pest. This product has been named Ledprona, and further products are expected to be commercialized against other pathogens [235].

## 4. Conclusions and Future Perspectives

The RNA-based strategies described here are excellent alternatives for plant disease control needed in the current era of food insecurity and climate change. To replace harmful chemicals with these tools in the field, further research is needed, focusing especially on the feasibility of their application, the efficacy against different pathogens, and the durability of the plant protection they provide. 

On the one hand, there has been an increasing interest in hypovirulent mycoviruses as potential biocontrol agents against fungi in recent years, but only CHV-1 (+ssRNA) [64] and SsHADV-1 (ssDNA) [10] have been proven effective in the field. Hence, basic research is needed in order to be able to predict which viruses are candidates to develop hypovirulence in pathogens, taking advantage of deep sequencing technologies, which are yielding huge amounts of new information on viral sequences. Although there are numerous mechanisms of action that lead to hypovirulence, basic research on this aspect is also important because studying these mechanisms may enable the development of technologies mimicking them and thus not limit the disease control to virus infections that are ineffective due to the horizontal transmission handicap.

On the other hand, the discovery and subsequent findings on RNAi have expanded our understanding of gene regulation, and it has been widely used as a research tool to elucidate the function of genes [120]. More than that, it has also opened the door to its use in gene therapy and disease control, not only in medicine, with the first RNAi therapeutics already approved, but also in agriculture and forestry. RNAi technologies offer the possibility of gene sequence-dependent disease control, which gives specificity to this methodology by targeting a specific pathogen. Moreover, these approaches can provide more environmentally friendly plant protection against pathogens without damaging ecosystems.

In crops and forests, biological control by RNA is a beneficial alternative to using fungicides or pesticides, but it must also be cost-effective and easy to apply, especially in forest disease management, where the cost-benefit balance is crucial owing to the narrow profit margin.

Insights reviewed here and future discoveries will provide opportunities to learn and improve these innovative approaches to plant protection. 

## Figures and Tables

**Table 1 ijms-23-09236-t001:** Frequency counts * of known viral families infecting major fungal classes, grouped by viral genome. Data source: ICTV Master Species List 2020.

Mycovirus		Fungal Host											
Genome	Viral Family	Agaricomycetes	Dothideomycetes	Eurotiomycetes	Glomeromycetes	Leotiomycetes	Pezizomycetes	Pucciniomycetes	Saccharomycetes	Sordariomycetes	Tremellomycetes	Ustilaginomycetes	NA **^1^
ssDNA(−)	Anelloviridae	0	0	0	0	0	0	0	0	8	0	0	0
dsRNA	Amalgaviridae	0	0	0	0	0	0	0	2	0	0	0	1
	Chrysoviridae	0	5	6	0	0	0	0	1	10	0	0	0
	Curvulaviridae	1	1	0	0	0	0	0	0	2	0	0	0
	Megabirnaviridae	0	0	0	0	1	0	0	0	2	0	0	0
	Partitiviridae	18	3	6	0	6	0	1	0	16	0	0	0
	Polymycoviridae	0	2	3	0	0	0	0	0	3	0	0	0
	Quadriviridae	0	0	0	0	0	0	0	0	1	0	0	0
	Reoviridae	0	0	0	0	1	0	0	0	3	0	0	0
	Totiviridae	2	6	3	1	11	1	1	4	18	2	1	0
ssRNA(+)	Alphaflexiviridae	0	0	0	0	2	0	0	0	0	0	0	0
	Barnaviridae	1	0	0	0	0	0	0	0	0	0	0	0
	Botourmiaviridae	1	0	0	0	3	0	0	0	2	0	0	0
	Deltaflexiviridae	0	0	0	0	2	0	0	0	1	0	0	0
	Endornaviridae	9	1	0	0	6	1	1	0	1	0	0	0
	Fusariviridae	0	2	2	0	4	0	0	0	4	0	0	0
	Gammaflexiviridae	0	0	0	0	1	0	0	0	0	0	0	0
	Hypoviridae	0	0	0	0	3	0	0	0	11	0	0	0
	Mitoviridae	4	2	0	11	13	1	5	0	12	0	0	0
	Narnaviridae	0	1	1	0	1	0	0	2	2	0	0	0
	Nodaviridae	0	0	0	0	0	0	0	1	0	0	0	0
	Secoviridae	0	0	0	0	0	0	0	0	1	0	0	0
	Tombusviridae	0	0	0	0	1	0	0	0	1	0	0	0
	Virgaviridae	0	1	0	0	0	0	0	0	0	0	0	0
ssRNA(−)	Aspiviridae	0	0	0	0	0	0	0	0	0	0	0	1
	Mymonaviridae	0	0	0	0	2	0	0	0	0	0	0	0
	Phenuiviridae	1	0	0	0	0	0	0	0	1	0	0	0
ssRNA-RT	Metaviridae	0	1	0	0	0	0	0	0	0	0	0	0
DNA	Genomoviridae	0	0	0	0	1	0	0	0	0	0	0	0
NA **^2^	NA **^2^	5	6	4	1	10	0	0	2	24	0	0	0

* Frequency counts: number of known mycoviruses of a virus family that infects a given fungal class. ** NA: not available, ^1^ mycoviruses that infect fungi, but the fungal class is unknown, ^2^ unknown mycoviruses infecting a fungal class.

**Table 3 ijms-23-09236-t003:** List of web-based algorithms for the design of siRNAs.

Tool	Use	Source	URL	Access
siDESIGN Center	siRNAs design	Dharmacon	https://horizondiscovery.com/en/ordering-and-calculation-tools/sidesign-center accessed on 16 August 2022	Free-access
DsiRNA	Custom design of Dicer-Substrate siRNA (DsiRNA)	IDT	https://www.eu.idtdna.com/site/order/designtool/index/DSIRNA_CUSTOM accessed on 16 August 2022	Free-access
BLOCK-iT™ RNAi Designer	Design of siRNAs, shRNAs, Stealth RNAi™ siRNAs and miR RNAs	ThermoFisher	https://rnaidesigner.thermofisher.com/rnaiexpress/ accessed on 16 August 2022	Free-access
Sfold web server	Prediction of RNA secondary structure	Ding et al., 2004 [190]	https://sfold.wadsworth.org/cgi-bin/index.pl accessed on 16 August 2022	Free-access
siRNA at Whitehead	siRNAs design	Whitehead Institute for Biomedical Research Yuan et al., 2004 [191]	http://sirna.wi.mit.edu/ accessed on 16 August 2022	Free-access
siMax siRNA design tool	siRNAs design	Eurofins	https://eurofinsgenomics.eu/en/dna-rna-oligonucleotides/oligo-tools/sirna-design-tool/ accessed on 16 August 2022	Free-access
OfftargetFinder	Off-target prediction	Good et al., 2016 [192]	no longer available	Free-access
siDirect	siRNAs design and off-target prediction	Naito et al., 2009 [193]	http://sidirect2.rnai.jp/ accessed on 16 August 2022	Free-access
si-Fi (siRNA-Finder)	RNAi design and off-target prediction	Luck et al., 2019 [194]	open-source (CC BY-SA license) desktop software	Free-access
RNAfold WebServer	Prediction of RNA secondary structure	Institute for Theoretical Chemistry (University of Vienna)	http://rna.tbi.univie.ac.at/cgi-bin/RNAWebSuite/RNAfold.cgi accessed on 16 August 2022	Free-access
RNAxs web server	siRNAs design	Theoretical Biochemistry Group (University of Vienna), Institute of Molecular Biotechnology (IMBA) of the Austrian Academy of Sciences, Max Perutz Labs Vienna	http://rna.tbi.univie.ac.at/cgi-bin/RNAxs/RNAxs.cgi accessed on 16 August 2022	Free-access
RNA plfold	Assess the mRNA target site accessibility	Theoretical Biochemistry Group (University of Vienna)	https://www.tbi.univie.ac.at/RNA/RNAplfold.1.html accessed on 16 August 2022	Free-access

## Data Availability

Not applicable.

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
