# Peer review of "New Insights on the Integrated Management of Plant Diseases by RNA Strategies: Mycoviruses and RNA Interference"

_ijms, 2022, doi:10.3390/ijms23169236_

Round 1
Reviewer 1 Report
2022-JULY-14
Reviewing opinion on:
Manuscript number: ijms-1810050
Title: New insights on the integrated management of plant diseases by RNA Strategies: mycoviruses and RNA interference
Authors: Irene Teresa Bocos-Asenjo *, Jonatan Niño-Sánchez, Julio Javier Diez
Journal: International Journal of Molecular Sciences
General comments:
The authors reviewed the use of mycoviruses as biocontrol means to protect plants from certain type of fungal pathogens; and reviewed the used of HIGS and SIGS to combat plant diseases. The main concern is the second part of the review on HIGS and SIGS. Lines 553-723 (3 pages and more) are all about SIGS. Only two paragraphs describe HIGS (Lines 378-404; Lines 488-513), and whether Lines 488-513 should be put under the subsection “Environmental RNAi” is still a point for discussion. Besides, HIGS involves GMO; and SIGS rather GMO-free. And in the section 3.3 (Lines 487-552), most examples are about controlling fungal pathogens.
Would it be possible to re-organize the second part? – For example, focus only control fungal pathogens and oomycete by HISG and SIGS? Or re-arrange the sections 3.2 and 3.3 based on HIGS and SIGS?
If in the second part the authors would like to review control different types of diseases (not only fungal ones) by HIGS and SIGS, are the authors sure all examples from the literature included? - If not, the authors should make a choice and focuses.
Detailed comments are listed below.
Full descriptions of given abbreviations should be unified throughout the text:
e.g., Lines 65, 67: full description of abbreviations HIGS and SIGS should follow the way written in Line 21.
The way of writing virus name should follow “How to write virus, species, and other taxa names” at
https://talk.ictvonline.org/information/w/faq/386/how-to-write-virus-species-and-other-taxa-names;
“A species name* is written in italics with the first word beginning with a capital letter.
Other words only begin with a capital if they are proper nouns (including host genus
names but not virus genus names**) or alphabetical identifiers. A species name should
not be abbreviated.
A virus name should never be italicized, even when it includes the name of a host
species or genus and should be written in lower case. This ensures that it is
distinguishable from a species name, which otherwise might be identical. The first
letters of words in a virus name, including the first word, should only begin with a capital
when these words are proper nouns (including host genus names but not virus genus
names) or start a sentence. Single letters in virus names, including alphanumerical
strain designations, may be capitalized. In most texts, virus names are used much more
frequently than species names and may, therefore, be abbreviated.”
And abbreviation of a virus name should be unified throughout the text.
e.g., Line 88-89: “SsHADV-1” was used in literature.
Line 97-98, should give citation.
Line 126, dsRNA: if it first time appear in the text, should give full description here.
Line 132: is “virus-derived small RNA” a new term? Or the authors mean “virus-derived small interfering RNAs (vsiRNAs)”?
Line 264: Table 1, since the internal horizonal lines are not visible in final form, in current form, the readers cannot distinguish the items in different columns belong to which horizonal line.
Also, it would be better if authors could add the host plant species that fungus affects according to the literature cited. The full name of the viruses should be given as footnote, the Table itself should be understandable without referring to the text.
Line 301, full description for RISC is “RNA-induced silencing complex” in literatures.
Line 297-302: “a double stranded RNA of 19-25 nt” should be siRNA duplexes, it differed from siRNA. Please check how to describe the siRNA biogenesis.
Line 313: “sRNA” or “siRNA”, please check.
Line 320-323: to separate as two sentences would be easier understood.
Line 325: citation [116] is about miRNA. The basic question is: the authors would like to review both miRNA and siRNA as biocontrol methods (if it is true based on literatures), or focus on siRNA only. siRNA and miRNA are both small RNA, they may have something in common, but they are different classes of small RNA. It cannot be written both in mixed way without underlining. The authors should either focus on one of them or underline both.
Line 346, are citations [127-129] about RNAi therapy against human disease? If so, these citations can be deleted. This review is focused on agricultural and forestry sciences.
Line 365: citation [117] is about miRNA. siRNA or miRNA or both? See previous comment.
Line 372, should add citation again after “…..infection”.
Line388: G and M should not be capitalized.
Line 421: in citation [169], miRNA is included. See previous comments.
Line 433: “from” should change as “by”?
Line 490: is HIGS under the “environmental RNAi”? see Lines 405-406 on definition of “environmental RNAi”.
Line 517-518: about “external sRNAs and dsRNAs”, in most cases in SIGS the authors mentioned dsRNAs were applied (dsRNAs here seem automatically mean long dsRNA?). What is the difference between the external use sRNAs and dsRNAs? If yes, should be underlined.

Reviewer 2 Report
Thank you for submitting your manuscript to IJMS. The current article reviewed systematically with many references as following title: New insights on the integrated management of plant diseases 2 by RNA Strategies: mycoviruses and RNA interference. My comments are as below:
1. Too long to be read, so please make compact. But in the case of data, please do not present it as supplemetary data, but showt them directly as data (Figures and/or Tables).
2. It would be good to focus on "Molecular Science" and deliver your manuscript concisely and clearly to the readers, so that it is appropriate for the aim/scope of this journal.
3. Please reduce your references section. Instead of reducing references, briefly describe future prospectives or directions for future progress.
Reviewer 3 Report
The manuscript by Bocos-Asenjo et al is a nice summary and overview on models and techniques for plant disease control by RNA strategies. Old and new publications are put together in the different chapters and the logic behind the presentation is easy to follow. Clearly an interesting review, in particular for those scientists working on other strategies of disease control in crops and model plants. Some chapters contain too many repetitions, as outlined below.
I have several minor points
The manuscript does not mention why mycoviruses can repress disease symptoms of a pathogenic fungus in plants. Is there anything known about the molecular mechanism? Maybe this can be explained for the reader. (l.117f, and following chapter). This chapter deals with sequences techniques and identification of the mycoviruses, but not really with their function. Als l. 226 and sentences before, or l.255. No idea about the action and no hypothesis for the mechanisms that can be experimentally tested?
The Abstract in l.12 mentions “hypovirulence” but it needs to be explained in the Abstract, since it is not clear for a general reader.
l. 107/8 (they are considered crytic) after “host”.
l. 109 Mycoviruses that host endophytic fungi?
Chapter 2.4 requirements for use as biocontrol agent.
Point (e) is very specific. While the other points are obvious, what is so specific for the host silencing RNA machinery? There are also other mechanisms of the host to control propagation of viruses in the fungal hosts. And if this is the main technique, described why.
RNAi Although transgenic, there might be RNAi crops line available with benefits for plant pathogen resistance which are used for application. The authors should separate better potential techniques tested in labs and RNAi tools already tested in the field or commercially available. Some of these lines are mentioned later.
l. 485ff: GreenLight Bioscience product. I assume that the reader wants to know more about it and how it differs from other approaches. There is more information in ref. 194, but also 193 which could be mentioned here.
l.488-513: maybe better to describe first HIGS, then SIGS. Or present the examples (also) in a table, separating HIGS and SIGS examples.
L.599 Instability of what RNA?
l. 595-621 (-638): mentions several times that specificity of the sequence choice is important, reduce repetitions in this section.
l. 651: loaded with zeta potential? do the authors mean measured with …. Or charged with?
l. 658: siRNAs
3.4.6. For agricultural application, this aspect is important. Although it is often a secret of how the companies design their products, the authors could potentially provide more available information which might be attractive for readers with interest in large scale applications.
Overall, the review presents a nice overview on RNA strategies, from their discoveries to (potential) applications. I propose to highlight the potential of this technology by mentioning examples which are already applied (e.g. in agriculture, forestry). (cf. chapter 3.4.6).
Round 2
Reviewer 1 Report
2022-AUG-04
Reviewing opinion on:
Manuscript number: ijms-1810050-revised
Title: New insights on the integrated management of plant diseases by RNA Strategies: mycoviruses and RNA interference
Authors: Irene Teresa Bocos-Asenjo *, Jonatan Niño-Sánchez, Julio Javier Diez
Journal: International Journal of Molecular Sciences
Comments:
The manuscript has been improved and re-organized.
Some minor comments are listed below.
Lines 66, 67: give full name for “GM” “GMO”, if first appear in the main text.
Latin names of species and “in vitro” “in vivo”: should be in italic: e.g., Lines 81, 93, 371, 413, 588, 593. Please check throughout the text.
Line 138, Table 1: footnote should be added to explain the meaning of “Frequency count”, “NA”. What dose it the values mean in the Table. Dose the value “5” mean 5 virus species within Chrysoviridae family?
Line 263: “-” should be deleted.
Line 289: should be “inoculation”
Line 421: “various fungal pathogens” ?
Line 464: which size range (below 800 bp) is the most effective?
Line 553: would “the genetically transformed vector” be treated as GM or not?
Reviewer 2 Report
Thank you for revising your manuscript. My comments are as follows:
1. My comments were fairly resolved.
2. By the way, one thing needs to be confirmed. Previously published papers of the corresponding author are not searched even if the papers appearing in References section are checked. "Review article" must consider the research achievements of the corresponding author or the first author, so I am sure that it will be of great help to the review if you present previously published papers which are involved in this title.
3. This manuscript is more suitable for the aim and scope of PLANT(Basel). It's just my opinion. Please consider that.
